# Influence of an External Electric Field and Dissipative Tunneling on Recombination Radiation in Quantum Dots

**DOI:** 10.3390/s22041300

**Published:** 2022-02-09

**Authors:** Vladimir D. Krevchik, Aleksei V. Razumov, Mikhail B. Semenov, Saygid U. Uvaysov, Vladimir P. Kulagin, Paweł Komada, Saule Smailova, Aisha Mussabekova

**Affiliations:** 1Faculty of Information Technology and Electronics, Penza State University, 440026 Penza, Russia; fpite@pnzgu.ru (V.D.K.); physics@pnzgu.ru (A.V.R.); 2MIREA—Russian Technological University, 119454 Moscow, Russia; uvajsov@mirea.ru (S.U.U.); kulagin@mirea.ru (V.P.K.); 3Department of Electronics and Information Technologies, Lublin University of Technology, Nadbystrzycka 38d, 20-618 Lublin, Poland; p.komada@pollub.pl; 4D. Serikbayev East Kazakhstan Technical University, 69 Protozanov Str., Ust-Kamenogorsk 070004, Kazakhstan; saule_smailova@mail.ru; 5Academy of Logistics and Transport, 97 Shevchenko Str., Almaty 050012, Kazakhstan; aisha_1292@mail.ru

**Keywords:** dissipative tunneling, spectral intensity of recombination radiation, quasi-stationary *A^+^*-state, quantum dot, impurity complexes

## Abstract

The effect of an external electric field and dissipative tunneling on the spectral intensity of recombination radiation in a quantum dot with an *A^+^ + e* impurity complex (a hole localized on a neutral acceptor interacting with an electron localized in the ground state of the quantum dot) is studied in the zero-radius potential model in the adiabatic approximation. The probability of dissipative tunneling of a hole is calculated in the one-instanton approximation. A high sensitivity of the recombination radiation intensity to the strength of the external electric field and to such parameters of the surrounding matrix (dissipative tunneling parameters) as temperature, the constant of interaction with the contact medium (or the heat-bath), and the frequency of phonon modes, has been revealed. It is shown that an external electric field leads to a shift of the recombination radiation threshold by several tens of meV, and a change in the parameters of dissipative tunneling has a noticeable effect on the spectral intensity of recombination radiation. It is shown that the resonant tunneling effect manifests itself in the form of “dips” in the field dependence of the spectral intensity of recombination radiation, which occur at certain values of the external electric field strength and temperature. This opens up certain prospects for the use of the considered system “quantum dot—impurity complex *A^+^ + e*” under conditions of dissipative tunneling for the study and diagnostics of biological objects.

## 1. Introduction

As is known, quantum technologies are currently a priority direction in the development of sensors. In particular, there are three groups of quantum sensors: “(1) clocks, gravimeters, gradiometers; (2) electric and magnetic field sensors; (3) sensors for quantum metrology”. It was noted that “… the IPS SB RAS has begun the development of single photon detectors based on avalanche photodiodes with heterostructures for fiber-optic quantum communications; at Moscow State University named by M.V. Lomonosov, in the framework of the direction of quantum metrology of the roadmap for quantum sensors, physical foundations have been developed and prototypes of devices for “absolute quantum photometry” have been built…; at the Physico-technical Institute named by A.F. Ioffe, RAS, the fundamental problem of detecting weak magnetic fields with nm—resolution has been solved. … Nanocrystals of a given polytype have been created, which can be combined with confocal and probe microscopy, resulting in atomic spatial resolution”. Research in this field, in particular, based on quantum dots with impurity complexes in the external control electric field, is undoubtedly a topical area of research in the field of modern optoelectronics. An important factor that significantly determines the optical response of quantum dots, in addition to the external electric field, is the influence of the surrounding matrix within the framework of the dissipative tunneling mechanism [1,2,3,4]. When discussing the effect of the surface on the photoluminescence of semiconductor quantum dots during the development of quantum sensors, it was noted [5] that it is important to take into account as role of recombination radiation as also tunneling mechanism. Among the possible mechanisms of the quantum dots interaction with the environment (with the heat-bath), we propose to take into account the mechanism of quantum tunneling with dissipation [1,2,3,4]. In addition, as model parameters of dissipative tunneling, it is necessary to take into account the temperature of the matrix surrounding the quantum dot with the impurity complex “*A^+^ + e*”) in the external control electric field, the parameter of interaction with the contact medium (with the heat-bath), frequencies of optical and acoustic phonon modes of the heat-bath matrix.

At present, interest is attracted by the effect of recombination radiation in the development of quantum sensors [5]. 

It is well known [6] that the core-shell quantum dots have a higher quantum yield of radiation than fluorescent chromophores, optical activity in the long-wavelength region of the spectrum, and significantly higher photochemical stability. Thus, these spectral properties of the core-shell quantum dots are very promising for research in biology and medicine and for the usage of the core-shell quantum dots as biological sensors. In [6], a new method for diagnosing amino acids (or other ligands) using quantum dots was proposed. We are talking about the effect of the interaction of amino acids (ligands) with quantum dots on the energy spectrum of quantum dots. This interaction affects the spectrum of radiative recombination of electrons and holes in quantum dots due to the different distribution of the field around them. Therefore, by changing the spectrum of radiative recombination of the core-shell quantum dots, it is possible to identify a biological object. 

There are well-known works where the usage of quantum dots as biomedical sensors has been investigated ([6,7]; see also the author’s patent [8]; [9,10]). Considering our work on the “Influence of an external electric field and dissipative tunneling on recombination radiation in quantum dots”, we also assume the usage of the proposed model for the purposes of modern nanomedicine and optoelectronics with controllable characteristics. 

The aim of this work is to show that the recombination radiation in the “quantum dot—impurity complex” system under the conditions of dissipative tunneling can be effectively used to determine parameters of the medium surrounding the quantum dot. This is important, for example, for nanomedicine, where the diagnosis of amino acids [6,7] and of oncological tumors [8] takes place, as well as for nanotechnology of quasi-zero-dimensional structures in cases where the surrounding matrix or the heat-bath can lead to a significant modification of the energy spectrum of the array of quantum dots due to tunneling transparency of potential barriers. Figure 1 shows the structure of the considered system “quantum dot—impurity complex”. 

Note that most of the currently existing works that take into account the tunneling mechanism in describing the optical properties of quantum dots offer only numerical estimates of the results obtained. One of the main advantages of this work is obtaining the main results in an analytical form. 

## 2. Materials and Methods

The theoretical consideration of the temperature effect on the energy levels in a semiconductor quantum dot (QD) was carried out by a statistical method, under assumption that the main contribution to the temperature dependence is made by the electron-phonon interaction. The dispersion equation, which determines the binding energy of a hole in an impurity complex $A^{+}+e$ in a spherically symmetric quantum dot, has been obtained in framework of the adiabatic approximation by the zero-range potential method. Calculation of the dissipative tunneling probability is performed in the one-instanton approximation. Calculation of the recombination radiation intensity in a quasi-zero-dimensional structure with impurity complexes is performed in the dipole approximation taking into account the radius dispersion of quantum dots and the Lorentz broadening of energy levels. The curves are plotted for the case of InSb quantum dots. In semiconductor nanostructures, the concept of deep and shallow impurities is relative since the depth of the impurity level depends on the characteristic size of the nanostructure. *A^+^*-centers appear due to the attachment of an additional hole to a neutral acceptor, and the interaction potential of a hole with a neutral acceptor is not Coulomb, but short-range. Such centers have been found in quantum wells (GaAs/AlGaAs) [11,12]. As is known, the effective mass approximation is applicable if the exciton Bohr radius $a_{ex}$ is large compared to the crystal lattice constant a. For a QD with a radius $R_{0}$, the applicability condition for the effective mass method is that it ($R_{0}$) must exceed the a value by several orders of magnitude. It is easy to show that this criterion is satisfied by semiconductors with a small effective electron mass. Thus, for *InSb*-based QDs with effective masses of electrons $m_{e}^{*}=0.013m_{e}$ and holes $m_{h}^{*}=0.6m_{e}$, the exciton Bohr radius is $a_{ex}\approx 70\;\mathrm{nm}$, which is two orders of magnitude larger than the lattice constant (a=0.65 nm). In this work, calculations and plotting are performed for the radius of the QD, in this case $N=\frac{R_{0}}{a}=108$. Thus, the number of atomic layers in an *InSb* crystal turns out to be sufficient for the applicability of the effective mass method. The value of *N* given can serve as an estimate for the number of unit cells of the material crystals, which are contained in the QD.

## 3. Results

### 3.1. Binding Energy of a Quasi-Stationary A^+^-State in a Quantum Dot in the Presence of Tunneling Decay in an External Electric Field

Let us consider *A^+^*-center in a QD, which can be formed due to the addition of an additional hole to a neutral acceptor. Exploring dependence of the impurity complex binding energy on such parameters of the matrix, surrounding the quantum dot, as temperature, the strength of the external electric field, the frequency of phonon modes, and the coupling constant with the contact medium (or with the heat-bath), is very important. It is also important to take into account the temperature dependence of the photoluminescence of the quantum dots under consideration [13,14,15,16,17]. As is known [18,19], for most semiconductor materials, the most significant contribution to the temperature dynamics of these QD’s energy levels is made by the electron—phonon interaction ([18,19]). A theoretical consideration of the temperature effect on the electronic energy levels in a semiconductor QD can be carried out by a statistical method (see, for example, [18]), under assumption that the main contribution to the temperature dependence is made by the electron-phonon interaction. The probability that an electron is in a state with energy *E_n_*, is given by the Fermi function
(1)$$f\left(\Psi \right)=\frac{1}{1+\exp\left[\frac{E_{n}-E_{F}}{kT}\right]}$$
where *E_F_*—is the Fermi energy, and the quantity *E_n_*, as is known, in quantum statistics has the meaning of free energy and is determined by the quantum canonical Gibbs distribution
(2)$$w\left(E\right)=\exp⟮\frac{E_{n}-E}{kT}⟯,$$
where *w*(*E*)—is the probability of a given discrete energy value *E*.

In our case, this is the electron energy *E_n_*, averaged over the vibrational states of the crystal lattice, defined as
(3)$$\exp⟮-\frac{E_{n}}{kT}⟯=\sum_{i=0}^{\infty }\exp⟮-\frac{E_{ni}}{kT}⟯,$$
here *E_ni_*—is the energy of an electron when it is in the *n*-th state, and the crystal lattice is in the *i*-th state. This energy is the sum of the electronic term *E_en_*, the phonon energy *E_p_*, and the energy of the electron-phonon interaction *E_ep_*: (4)$$E_{ni}=E_{en}+E_{p}+E_{ep}.$$

If **q**—is the phonon wave vector, $\omega_{LA}\left(q\right),\omega_{TA}\left(q\right)$—frequencies of the longitudinal (*LA*) and transverse (*TA*) acoustic phonons, and $\hbar \omega_{LA}\left(q\right),\hbar \omega_{TA}\left(q\right)$—energies of the two-particle electron-phonon interaction, then for *E_p_* and *E_ep_* we can write
(5)$$E_{p}+E_{ep}=\int_{-\pi /a}^{\pi /a}\hbar \left[\omega_{LA}\left(q\right)+2\omega_{TA}\left(q\right)+\omega_{eLA}\left(q\right)+2\omega_{eTA}\left(q\right)\right]\left[N_{i}+\frac{1}{2}\right]dq,$$
here *a*—is the lattice constant, $N_{i}=0,1,2,\ldots$—phonon occupation numbers.

Integration in (5) requires knowledge of the corresponding dispersion laws. For three branches of acoustic phonons (longitudinal and two transversal), in the long-wave approximation, the linear dispersion law is valid
(6)$$\omega_{p}\left(q\right)=v_{j}q,$$
where $v_{j}$—is the sound velocity for the *j*-th phonon branch. 

The energy of the electron-phonon interaction is determined by the expression [20]
(7)$$\hbar \omega_{ep}\left(q\right)=\frac{\Xi }{v_{j}}\sqrt{\frac{kT}{2\rho V}}G,$$
where Ξ—is the deformation potential, ρ—the density of the QD material, G—the overlap integral.

Then, performing integration in (5), we obtain
(8)$$\exp⟮-\frac{E_{n}}{kT}⟯=\exp⟮-\frac{E_{en}}{kT}⟯\sum_{i=0}^{\infty }\exp⟮-\frac{\Omega }{\sqrt{kT}}⟮\frac{1}{v_{LA}}+\frac{2}{v_{TA}}⟯⟮N_{i}+\frac{1}{2}⟯⟯,$$
here, $v_{LA}$ and $v_{TA}$ are both the velocities of the longitudinal and transversal phonons and the next notation has been introduced
(9)$$\Omega =\frac{\pi \hbar \Xi G}{a}\sqrt{\frac{2}{\rho V}}.$$

Summing up in (8) for the temperature dependence of the *n*-th energy level, we obtain the expression
(10)$$E_{n}=E_{en}+kT\ln\left[4sh⟮\frac{\Omega }{v_{LA}\sqrt{kT}}⟯sh⟮\frac{2\Omega }{v_{TA}\sqrt{kT}}⟯\right].$$

As it has been mentioned above, it is important to take into account the temperature dependence of the photoluminescence of the quantum dots under consideration [13,14,15,16,17,20,21]. Let us further consider problem of the hole quasistationary states in an impurity complex $A^{+}+e$ in the semiconductor spherically symmetric QD. The interaction of an electron in the ground state in a QD with a hole localized at the $A^{0}$-center will be considered in framework of the adiabatic approximation [22]. In this case, the electron potential $V_{n,l,m}\left(\vec{r}\right)$, acting on the hole can be considered as averaged over the electron motion [22]
(11)$$V_{n,l,m}\left(\vec{r}\right)=-\frac{e^{2}}{4\pi \varepsilon_{0}\varepsilon }\int_{0}^{R_{0}}\frac{{\left|\Psi_{n,l,m}\left(\vec{r}\right)\right|}^{2}}{\left|\vec{r}-{\vec{r}}_{e}\right|}d{\vec{r}}_{e},$$
where *e* is the electron charge; *ε* is the dielectric constant of the QD material; *ε*_0_ is the electrical constant; $\Psi_{n,l,m}\left(\vec{r}\right)$ is the wave function of the QD electron; *m* = 0, ±1, ±2, … is the magnetic quantum number; *l* = 0, 1, 2, … is the orbital quantum number.

In the first order of the perturbation theory for the electron ground state (*m* = 0, *l* = 0) potential (11) can be written in the form
(12)$$V_{n,0,0}\left(r\right)=-\frac{e^{2}\beta_{n}}{4\pi \varepsilon_{0}\varepsilon R_{0}}+\frac{m_{h}^{*}\omega_{n}^{2}r^{2}}{2},$$
where $\beta_{n}=\gamma_{0}-Ci\left(2\pi n\right)+\ln\left(2\pi n\right)$; $\hbar \omega_{n}=\sqrt{2\hbar^{2}\pi^{2}n^{2}e^{2}/3m_{h}^{*}R_{0}^{3}4\pi \varepsilon_{0}\varepsilon }$; $\gamma_{0}=1.781$ is the Euler’s constant; Ci(x) denotes the integral cosine; *n* is the electron radial quantum number; $m_{h}^{*}$ is the hole effective mass. 

Since the confining potential of a QD, generally speaking, should have a finite depth, then in our model of the hole confinement potential (12), the potential amplitude *U*_0_ is an empirical parameter and satisfies the relation $U_{0}=-e^{2}\beta_{n}/4\pi \varepsilon_{0}\varepsilon R_{0}+m_{h}^{*}\omega_{n}^{2}R_{0}^{2}/2=m_{h}^{*}\omega_{0}^{2}R_{0}^{2}/2$, whence $\omega_{0}=\sqrt{\omega_{n}^{2}-e^{2}\beta_{n}/2\pi \varepsilon_{0}\varepsilon m_{h}R_{0}^{3}}$ is the characteristic frequency of the hole confining potential of a QD within the adiabatic approximation, in this case $U_{0}/\left(\hbar \;\omega_{\;0}\right)>>1$.

Usage of the adiabatic approximation makes it possible to take into account the effect of an external electric field on the hole bound states. Let the electric field strength ${\vec{E}}_{0}$ be directed along the *x* coordinate axis, then the oscillator energy levels (12), taking into account (10), are given in the form
(13)$$E_{n_{1},n_{2},n_{3}}^{n,\;0,\;0}\left(T\right)=-\frac{e^{2}}{\varepsilon R_{0}}\beta_{h}-\frac{{\left|e\right|}^{\;2}\;E_{0}^{2}}{2\;m_{h}\;\omega_{\;n}^{\;2}}+\hbar \omega_{n}⟮n_{1}+n_{2}+n_{3}+\frac{3}{2}⟯+kT\ln\left[4sh⟮\frac{\Omega }{v_{LA}\sqrt{kT}}⟯sh⟮\frac{2\Omega }{v_{TA}\sqrt{kT}}⟯\right]$$
and corresponding the one-particle wave functions are written as
(14)$$\Psi_{n_{1},\;n_{2},\;n_{3}}^{n}\left(x,y,z\right)=C_{n}\exp⟮-\frac{{\left(x-x_{0}\right)}^{2}+y^{2}+z^{2}}{2a_{n}^{2}}⟯H_{n_{1}}⟮\frac{x-x_{0}}{a_{n}}⟯H_{n_{2}}⟮\frac{y}{a_{n}}⟯H_{n_{3}}⟮\frac{z}{a_{n}}⟯,$$
where $C_{n}={\left[2^{n1+n2+n3}n_{1}!n_{2}!n_{3}!\pi^{3/2}a_{n}^{3}\right]}^{-1/2}$; $a_{n}=\sqrt{\hbar /\left(m_{h}^{*}\omega_{n}\right)}$; $x_{0}=\left|e\right|\;E_{0}/\left(m_{h}^{*}\;\omega_{n}^{2}\right)$; $H_{n}\left(x\right)$—Hermite polynomials; $n_{1}$,$n_{2}$,$n_{3}$—are the quantum numbers corresponding to the energy levels of the harmonic oscillator.

We will assume that the decay process of the quasi-stationary level of the $A^{+}$-center is due to dissipative tunneling.

The quantum tunneling with dissipation theory was actively developed in the early 1980-th in the works of A. I. Larkin et al. [1]. Interest in this theory was caused by the fact that for the case of a model potential of the “cubic parabola” type, describing Josephson contacts, for the case of a finite temperature, taking into account the heat-bath influence in the semiclassical instanton approximation, it was possible to obtain analytical results for the tunneling probability. The case of sufficiently low temperatures was considered in the works of A. J. Leggett, and at arbitrary temperatures and “viscosity coefficients” in the works of A. I. Larkin, Yu. N. Ovchinnikov and other authors [1]. In this case, the effects of dissipative tunneling were manifested in sufficiently small contacts, when, under certain conditions, voltage surges appeared [1]. An analogue of the Josephson effect can manifest itself in nuclear transformations, in which case the fission of atomic nuclei is interpreted as collective quantum tunneling with dissipation. In this case, role of the collective coordinate q is played by the quadrupole moment of the nucleus or its shape, and role of the heat-bath is played by single-nucleon excitations. Somewhat later, using the formalism developed in the works of A. I. Larkin, Yu. N. Ovchinnikov, B. I. Ivlev, A. J. Leggett, theory of quantum tunneling with dissipation for the case of 1D and 2D oscillator potentials was developed by A. A. Ovchinnikov, Yu. I. Dakhnovsky, V. A. Bendersky, E. I. Katz, et al. first to describe the kinetics of low-temperature adiabatic chemical reactions as tunnel systems with dissipation. Subsequently, the instanton method was generalized to the case of tunnel-coupled nanostructures. In particular, by combining the instanton method with the zero-range potential method, it was possible to obtain important analytical results describing the optical and transport properties of quantum molecules with impurity centers [1].

At the same time, a significant conclusion of the research was the statement that the above-mentioned optical and transport properties are significantly influenced not only by the dimension, size, shape of nanostructures, the presence of impurities, external fields, and final temperature but also by the thermostat matrix (or by the heat-bath) itself, in which these nanostructures are synthesized. In [1], when considering the experimentally observed 1D and 2D effects of dissipative tunneling within the limits of weak and strong dissipation, the field and temperature dependences of these effects were studied theoretically and experimentally, demonstrating their controllability. 

Thus, for metallic (golden and zirconium) quantum dots synthesized in a dielectric matrix, in a system of a combined atomic force and scanning tunneling microscope at a certain value of the external electric field strength, when the model double-well 1D-oscillator potential becomes symmetric, in the limit of weak dissipation in the field dependence of the tunneling probability, a single temperature-dependent peak is observed [1]. In the case of planar structures with synthesized QDs from colloidal gold, the field dependence of 2D dissipative tunneling can exhibit single and double bifurcations in the form of characteristic kinks, in the vicinity of which quantum beat regimes are realized [3]. In the strong dissipation limit for single InAs QDs, taking into account the influence of two local phonon modes on the field dependence of the probability of 1D dissipative tunneling, an oscillating mode with a nonequidistant spectrum of peaks was observed, which qualitatively coincides with the experimental I–V characteristic [2]. For structures with tunneling photodiodes based on tunneling coupled asymmetric InAs/GaAs quantum molecules, the experimental dependence of the photoconductivity qualitatively corresponded to the field dependence of the probability of 1D dissipative tunneling taking into account the influence of two local phonon modes (longitudinal and transversal optical phonons) [4].

It should be noted that in the one-instanton approximation, the decay probability (of dissipative tunneling), under conditions of an external electric field, can be represented in the form $\Gamma_{0}=B\exp\left(-S\right)$ (see also Appendix A), where S and B are determined as (in Bohr units) [1]: (15)S=12⟮b′0+x0∗a′0+x0∗+1⟯ ⟮3−b′0+x0∗a′0+x0∗⟯ τ−0∗12 β∗⟮b′0+x0∗a′0+x0∗+1⟯2τ20∗−12γ∗⟮b′0+x0∗a′0+x0∗+1⟯2×⟮(1−x2∗)x1∗[cth(β∗x1∗)−1sh(β∗x1∗)(ch[(β∗−τ0∗)x1∗]−ch[β∗x1∗])+ch((β∗−τ0∗)x1∗)]−(1−x1∗)x2∗[cth(β∗x2∗)−1sh(β∗x2∗)(ch((β∗−τ0∗)x2∗)−ch(β∗x2∗))+ch((β∗−τ0∗)x2∗)]⟯
(16)$$\begin{array}{ll} B= & \frac{2E_{d}\sqrt{U_{0}^{*}}}{\hbar \;\sqrt{\pi }}\left(\frac{{b^{\prime }}_{0}+x_{0}^{*}}{{a^{\prime }}_{0}+x_{0}^{*}}+1\right)\sqrt{\varepsilon_{T}^{*}}\times \\ & \left\{A^{*}\left[\beta_{1}^{*}ch\left(\frac{\beta_{1}^{*}}{2}\right)-1\right]+D^{*}\left[\beta_{2}^{*}ch\left(\frac{\beta_{2}^{*}}{2}\right)-1\right]+A^{*}\left(1-\frac{\beta_{1}^{*}}{2}\frac{ch\left(\frac{\beta_{1}^{*}}{2}-\tau_{01}{}^{*}\right)}{sh\left(\frac{\beta_{1}^{*}}{2}\right)}\right)+D^{*}\left(\frac{\beta_{2}^{*}}{2}\frac{ch\left(\frac{\beta_{2}^{*}}{2}-\tau_{02}{}^{*}\right)}{sh\left(\frac{\beta_{2}^{*}}{2}\right)}-1\right)\right\}\times \\ & {\left(A^{*}\left[\frac{\beta_{1}^{*}}{2}\frac{ch\left(\frac{\beta_{1}^{*}}{2}-\tau_{01}{}^{*}\right)}{sh\left(\frac{\beta_{1}^{*}}{2}\right)}-1\right]+D^{*}\left[\frac{\beta_{2}^{*}}{2}\frac{ch\left(\frac{\beta_{2}^{*}}{2}-\tau_{02}{}^{*}\right)}{sh\left(\frac{\beta_{2}^{*}}{2}\right)}-1\right]\right)}^{-\frac{1}{2}} \end{array}$$
where
$x_{1,2}^{*}=\frac{1}{2}\left[\frac{\varepsilon_{L}^{*2}a^{*2}}{4U_{0}^{*}}+1+\frac{\varepsilon_{c}^{4}a^{*2}}{4\varepsilon_{L}^{*2}U_{0}^{*}}\mp \sqrt{{\left(\frac{\varepsilon_{L}^{*2}a^{*2}}{4U_{0}^{*}}+1+\frac{\varepsilon_{c}^{4}a^{*2}}{4\varepsilon_{L}^{*2}U_{0}^{*}}\right)}^{2}-\frac{\varepsilon_{L}^{*2}a^{*2}}{U_{0}^{*}}}\right]$,$\gamma^{*}=\sqrt{{\left(\varepsilon_{L}^{*2}a^{*2}/\left(4\;U_{0}^{*}\right)+1+\varepsilon_{c}^{*\;4}a^{*2}/\left(4\;\varepsilon_{L}^{*2}U_{0}^{*}\right)\right)}^{2}-\varepsilon_{L}^{*2}a^{*2}/U_{0}^{*}}$, $\tau_{0}^{*}=arcsh\left[⟮1-\frac{{b^{\prime }}_{0}+x_{0}^{*}}{{a^{\prime }}_{0}+x_{0}^{*}}⟯sh\left(\beta^{*}\right)/⟮1+\frac{{b^{\prime }}_{0}+x_{0}^{*}}{{a^{\prime }}_{0}+x_{0}^{*}}⟯\right]+\beta_{T}^{*}$,$\varepsilon_{T}^{*}=kT/E_{h},\;\;\varepsilon_{L}^{*}=\hbar \omega_{L}/E_{h}$, $\varepsilon_{c}^{*}=\hbar \sqrt{c}/E_{d}$, $\beta_{T}^{*}=\sqrt{U_{0}^{*}}/a^{*}\varepsilon_{T}^{*}$, ${b^{\prime }}_{0}=b_{0}/a_{h}$, ${a^{\prime }}_{0}=a_{0}/a_{h}$, $x_{\;0}{}^{*}=x_{0}/a_{h}$; $E_{h}\;\mathrm{and}\;a_{h}$ are the Bohr energy and hole radius, respectively;$A^{*}=\left(2\;\varepsilon_{L}^{*2}a^{*2}-x_{1}^{*}\right)/\left(\left(x_{1}^{*}-x_{2}^{*}\right)x_{1}^{*}\right)$, $D^{*}=\left(2\;\varepsilon_{L}^{*2}a^{*2}-x_{2}^{*}\right)/\left(\left(x_{1}^{*}-x_{2}^{*}\right)x_{2}^{*}\right)$, $\beta_{1}^{*}=\sqrt{2\;U_{0}^{*}x_{1}^{*}}/\left(a^{*}\varepsilon_{T}^{*}\right)$, $\beta_{2}^{*}=\sqrt{2\;U_{0}^{*}x_{2}^{*}}/\left(a^{*}\varepsilon_{T}^{*}\right)$, $\tau_{01}^{*}=\tau_{0}^{*}\sqrt{x_{1}^{*}}/\sqrt{2}$, $\tau_{02}^{*}=\tau_{0}^{*}\sqrt{x_{2}^{*}}/\sqrt{2}$.

Using the procedure of the zero-range potential method (see, for example, [23]), we obtain an equation that determines the hole energy dependence in the complex $A^{+}+e$ on temperature *T*, QD parameters, and on dissipative tunneling. The short-range potential of an impurity is modeled by the zero—range potential with intensity $\gamma =2\pi \hbar^{2}/(\alpha m_{h}^{*})$, which has the form: (17)$$V_{\delta }\left(x,y,z;x_{a},y_{a},z_{a}\right)=\gamma \delta \left(x-x_{a}\right)\delta \left(y-y_{a}\right)\delta \left(z-z_{a}\right)\left[1+\left(r-r_{a}\right)\frac{\partial }{\partial r}\right],$$
where α is determined by the binding energy $E_{i}$ of the same $A^{+}$-center in a bulk semiconductor.

In the effective mass approximation, the wave function $\Psi_{\lambda h}\left(x,y,z;x_{a},y_{a},z_{a}\right)$ of an electron localized at a short-range potential satisfies the Schrödinger equation
(18)$$\left(E_{\lambda h}-H\right)\;\Psi_{\lambda h}\left(x,y,z;x_{a},y_{a},z_{a}\right)=V_{\delta }\left(x,y,z;x_{a},y_{a},z_{a}\right)\Psi_{\lambda h}\left(x,y,z;x_{a},y_{a},z_{a}\right),$$
where $E_{{}_{h}}^{QD}=-\hbar^{2}\lambda^{2}/\left(2m_{h}^{*}\right)$—the eigenvalues of the Hamilton operator $H^{\delta }=H+V_{\delta }\left(x,y,z;x_{a},y_{a},z_{a}\right)$; $H=-\;\hbar^{2}/\left(2\;m_{h}^{*}\right)\;\nabla^{\;2}+m_{h}^{*}\;\omega_{0}^{2}\;\left(x^{\;2}+y^{\;2}+z^{\;2}\right)/2-\left|e\right|\;E_{0}\;x$.

To determine the binding energy of a hole in a complex $A^{+}+e$ in the adiabatic approximation, it is necessary to construct a one-particle Green’s function $G\left(x,y,z;x_{a},y_{a},z_{a};E_{\lambda n}\right)$ to the Schrödinger equation with a Hamiltonian, containing potential (17)
(19)$$G\left(x,y,z;x_{a},y_{a},z_{a};E_{\lambda h}\right)=-\sum_{n1,n2,n3}\frac{\Psi_{n_{1},\;n_{2},\;n_{3}}^{n*}\left(x_{a},y_{a},z_{a}\right)\Psi_{n_{1},\;n_{2},\;n_{3}}^{n}\left(x,y,z\right)}{-E_{{}_{h}}^{QD}+i\hbar \Gamma_{0}+E_{n_{1},\;n_{2},\;n_{3}}^{n,\;0,\;0}\left(T\right)},$$

The Lippmann-Schwinger equation for a $A^{+}$-state in a QD with a parabolic confinement potential can be written as
(20)$$\Psi_{h}\left(x,y,z;x_{a},y_{a},z_{a}\right)=\int_{-\infty }^{+\infty }\int_{-\infty }^{+\infty }\int_{-\infty }^{+\infty }dx_{1}dy_{1}dz_{1}G\left(x,y,z;x_{a},y_{a},z_{a};E_{\lambda h}\right)\times V_{\delta }\left(x,y,z;x_{a},y_{a},z_{a}\right)\Psi_{\lambda h}\left(x_{1},y_{1},z_{1},x_{a},y_{a},z_{a}\right)$$

Substituting (17) into (20), we obtain
(21)$$\Psi_{h}\left(x,y,z;x_{a},y_{a},z_{a}\right)=\gamma G\left(x,y,z;x_{a},y_{a},z_{a};E_{\lambda h}\right)\left(T\Psi_{h}\right)\left(x,y,z;x_{a},y_{a},z_{a}\right),$$
where
(22)$$\left(T\Psi_{\lambda h}\right)\left(x,y,z;x_{a},y_{a},z_{a}\right)\equiv \underset{\begin{array}{l} r\to r_{a}\\ \varphi \to \varphi_{a}\\ \theta \to \theta_{a} \end{array}}{\lim}\left[1+(r-r_{a})\frac{\partial }{\partial r}\right]\Psi_{\lambda h}\left(x,y,z;x_{a},y_{a},z_{a}\right)$$

Acting as an operator T on both sides of relation (21), we obtain an equation that determines dependence of the bound state energy $E_{\lambda h}$ of the $A^{+}$-center on the QD parameters, on the position ${\vec{R}}_{\;a}=\left(x_{a},y_{a},z_{a}\right)$ of the impurity, and on the temperature *T*:(23)$$\alpha =\frac{2\pi \hbar^{2}}{m_{h}^{*}}\left(TG\right)\left(x_{a},y_{a},z_{a};x_{a},y_{a},z_{a};E_{\lambda h}\right),$$
here α is determined by the energy $E_{\;i}$ of the bound state of the same $A^{+}$—the center in a massive semiconductor.

Then, for the Green’s function (19), taking into account (13) and (14), we obtain in the Bohr units
(24)$$\begin{array}{ll} G\left(x,y,z,x_{a},y_{a},z_{a};\eta_{\lambda h}^{2}\right)= & -\frac{1}{\pi^{\frac{3}{2}}a_{n}^{2}E_{h}}\exp⟮-\frac{{\left(x^{*}-x_{0}^{*}\right)}^{2}+y^{*2}+z^{*2}+{\left(x_{a}^{*}-x_{0}^{*}\right)}^{2}+y_{a}^{*2}+z_{a}^{*2}}{2}⟯\;\times \\ & \sum_{n_{1,}n_{2,}n_{3}}\frac{H_{n_{1}}\left(x^{*}-x_{0}^{*}\right)H_{n_{1}}\left(x_{a}^{*}-x_{0}^{*}\right)}{2^{n_{1}}!n_{1}!}\frac{H_{n_{2}}\left(y^{*}\right)H_{n_{2}}\left(y_{a}^{*}\right)}{2^{n_{2}}!n_{2}!}\frac{H_{n_{3}}\left(z^{*}\right)H_{n_{3}}\left(z_{a}^{*}\right)}{2^{n_{3}}!n_{3}!}\;\cdot \\ & {\left\{-\eta_{h}^{2}-\beta_{h}^{*}-\frac{x_{0}^{*2}}{4\beta^{-2}}+i4\Gamma_{0}^{*}+\beta^{-1}⟮n_{1}+n_{2}+n_{3}+\frac{3}{2}⟯+\frac{kT}{E_{h}}\ln\left[4sh⟮\frac{\Omega }{v_{LA}\sqrt{kT}}⟯sh⟮\frac{2\Omega }{v_{TA}\sqrt{kT}}⟯\right]\right\}}^{-1}, \end{array}$$
where the next notations are introduced $\eta_{h}^{2}=E_{{}_{h}}^{QD}/E_{h}$; $R_{0}^{*}=R_{0}/a_{h}$; $\beta =E_{h}/\hbar \omega_{n}$; $a_{n}^{*}=a_{n}/a_{h}$; $\beta_{h}^{*}=e^{2}\beta_{h}/\varepsilon R_{0}^{*}E_{h}a_{h}$; $\Gamma_{0}^{*}=\hbar \Gamma_{0}/4E_{h}$.

Further, given that
(25)$$\begin{array}{l} {\left(-\eta_{h}^{2}-\beta_{h}^{*}-\frac{x_{0}^{*2}\beta^{-2}}{4a_{h}^{*2}}+i4\Gamma_{0}^{*}+\frac{kT}{E_{h}}\ln\left[4sh\left(\frac{\Omega }{v_{LA}\sqrt{kT}}\right)sh\left(\frac{2\Omega }{v_{TA}\sqrt{kT}}\right)\right]+n_{1}+n_{2}+n_{3}+\frac{3}{2}\right)}^{-1}=\\ \beta^{-1}\int_{0}^{\infty }dt\exp\left[-t(-\beta \left(\eta_{h}^{2}+\beta_{h}^{*}+\frac{x_{0}^{*2}\beta^{-2}}{4a_{h}^{*2}}-i4\Gamma_{0}^{*}\right)+n_{1}+n_{2}+n_{3}+\frac{3}{2}+\frac{kT\beta }{E_{h}}\ln\left[4sh\left(\frac{\Omega }{v_{LA}\sqrt{kT}}\right)sh\left(\frac{2\Omega }{v_{TA}\sqrt{kT}}\right)\right]\right] \end{array}$$
expression for the Green’s function takes the following form
(26)$$\begin{array}{l} G\left(x,y,z,x_{a},y_{a},z_{a},\eta_{h}^{2}\right)=-\frac{1}{\pi^{\frac{3}{2}}a_{n}^{2}\beta E_{h}}\exp⟮-\frac{{\left(x^{*}-x_{0}^{*}\right)}^{2}+{\left(x_{a}^{*}-x_{0}^{*}\right)}^{2}+y^{2}+y_{a}^{2}+z^{2}+z_{a}^{2}}{2a_{n}}⟯\times \\ \int_{0}^{\infty }dt\exp[-t⟮-\beta ⟮\eta_{h}^{2}+\beta_{h}^{*}+\frac{x_{0}^{*2}\beta^{-2}}{4a_{h}^{*2}}-i4\Gamma_{0}^{*}⟯+\frac{kT\beta }{E_{h}}\ln\left[4sh⟮\frac{\Omega }{v_{LA}\sqrt{kT}}⟯sh⟮\frac{2\Omega }{v_{TA}\sqrt{kT}}⟯\right]+\frac{3}{2})⟯\times \;\\ \sum_{n_{1}=0}^{\infty }{⟮\frac{e^{-t}}{2}⟯}^{n_{1}}\frac{H_{n_{1}}⟮\frac{x-x_{0}}{a_{n}}⟯H_{n_{1}}⟮\frac{x_{a}-x_{0}}{a_{n}}⟯}{n_{1}!}\sum_{n_{2}=0}^{\infty }{⟮\frac{e^{-t}}{2}⟯}^{n_{2}}\frac{H_{n_{2}}⟮\frac{y}{a_{n}}⟯H_{n_{2}}⟮\frac{y_{a}}{a_{n}}⟯}{n_{2}!}\times \sum_{n_{3}=0}^{\infty }{⟮\frac{e^{-t}}{2}⟯}^{n_{3}}\frac{H_{n_{3}}⟮\frac{z}{a_{n}}⟯H_{n_{3}}⟮\frac{z_{a}}{a_{n}}⟯}{n_{3}!}. \end{array}$$

Taking summation over quantum numbers $n_{1},n_{2},n_{3}$ and separating the divergent part of the expression for the Green’s function (26), we obtain
(27)$$\begin{array}{ll} G\left(r,R_{a},\eta_{h}^{2}\right)= & -\frac{1}{\pi^{\frac{3}{2}}a_{n}^{2}\beta E_{h}}\int_{0}^{\infty }dt\exp[-t⟮-\beta ⟮\eta_{h}^{2}+\beta_{h}^{*}+\frac{x_{0}^{*2}\beta^{-2}}{4a_{h}^{*2}}-i4\Gamma_{0}^{*}⟯+\frac{3}{2}\;+\frac{kT\beta }{E_{h}}\ln\left[4sh⟮\frac{\Omega }{v_{LA}\sqrt{kT}}⟯sh⟮\frac{2\Omega }{v_{TA}\sqrt{kT}}⟯\right]⟯]\times \\ & \{{\left(1-\exp\left(-t\right)\right)}^{-\frac{3}{2}}\exp⟮-\frac{r^{2}+R_{a}^{2}}{2a_{n}}⟯\exp⟮\frac{2\left(rR_{a}\right)e^{-t}-\left(r^{2}+R_{a}^{2}\right)e^{-2t}}{a_{n}\left(1-e^{-2t}\right)}⟯-\frac{1}{t\sqrt{t}}\exp\left[-\frac{{\left(r-R_{a}\right)}^{2}}{2a_{n}^{2}t}\right]\}-\\ & \frac{\exp⟮-\sqrt{2⟮-\eta_{h}^{2}-\beta_{h}^{*}-\frac{x_{0}^{*2}\beta^{-2}}{4a_{h}^{*2}}+i4\Gamma_{0}^{*}+\frac{kT}{E_{h}}\ln\left[4sh⟮\frac{\Omega }{v_{LA}\sqrt{kT}}⟯sh⟮\frac{2\Omega }{v_{TA}\sqrt{kT}}⟯\right]+\frac{3}{2}⟯}\frac{\left|r-R_{a}\right|}{a_{n}}⟯}{2\pi^{\frac{3}{2}}a_{n}\beta E_{h}\left|r-R_{a}\right|} \end{array}$$

Substituting (27) into (23), we obtain the dispersion equation that determines dependence of the binding energy $E_{{}_{h}}^{QD}$ of a hole in the $A^{+}+e$ complex on the QD parameters, on temperature *T*, and on the electron quantum number *n*
(28)$$\begin{array}{ll} \eta_{i}= & \sqrt{-\eta_{h}^{2}-\beta_{h}^{*}-\frac{x_{0}^{*2}\beta^{-2}}{4a_{h}^{*2}}+i4\Gamma_{0}^{*}+\frac{kT}{E_{h}}\ln\left[4sh⟮\frac{\Omega }{v_{LA}\sqrt{kT}}⟯sh⟮\frac{2\Omega }{v_{TA}\sqrt{kT}}⟯\right]+\frac{3}{2}}+\\ & \sqrt{\frac{2}{\beta \pi }}\int_{0}^{\infty }dt\exp\{-t\beta ⟮-\eta_{h}^{2}-\beta_{h}^{*}-\frac{x_{0}^{*2}\beta^{-2}}{4a_{h}^{*2}}+i4\Gamma_{0}^{*}+\frac{3}{2\beta }+\frac{kT}{E_{h}}\times \\ & \ln\left[4sh⟮\frac{\Omega }{v_{LA}\sqrt{kT}}⟯sh⟮\frac{2\Omega }{v_{TA}\sqrt{kT}}⟯\right]⟯\}\left[\frac{1}{2t\sqrt{2t}}-{\left(1-e^{-2t}\right)}^{-\frac{3}{2}}\exp⟮-\frac{R_{a}^{{*}^{2}}\beta^{-1}}{2}\times \frac{1-e^{-t}}{1+e^{-t}}⟯\right] \end{array}$$
where $R_{0}^{*}=R_{0}/a_{h}$; $\eta_{i}=\sqrt{\left|E_{i}\right|/E_{h}}$.

It should be noted that the hole binding energy in the considered case is a complex quantity. Its real part determines the average binding energy of the resonance state of the $A^{+}$-center, EhQD¯=ReEhQD, and its doubled imaginary part determines the broadening of the corresponding energy level ΔEh=2ImEhQD. Figure 2a–c shows the result of a numerical analysis of the dispersion Equation (28) for the case of a centered $A^{+}$-center ($R_{a}^{*}=0$) at different values of the QD radius $R_{0}^{*}$. It was taken into account that the binding energy of the $A^{+}$-state is measured from the ground state level of the adiabatic oscillatory well.

As it can be seen from Figure 2a–c, that in the field dependence of the binding energy of the *A^+^*-state, there are “dips” at a fixed temperature. This is due to the “tuning” effect of the starting energy level of the quasistationary *A^+^*-state to the states, caused by the electron—phonon interaction in the matrix, surrounding the quantum dot, i.e., with the effect of resonant tunneling. The dip depth increases with increasing temperature, which is due to dynamics of the temperature-dependent peak in the field dependence of the dissipative tunneling probability [1].

A decrease in the binding energy of the *A^+^*-state with an increase in the strength of the external electric field is associated with the Stark shift in energy and with the polarization of the *A^+^*-center, and with an increase in temperature—with broadening of energy levels and the corresponding “spreading” of the wave function of the *A^+^*-state. One can also see (see Figure 2b,c) a sufficiently high sensitivity of the binding energy of the *A^+^*-state to the frequency of the phonon mode (Figure 1b) and to the constant of interaction with the contact medium (Figure 2c). An increase in the latter constant leads to blocking of tunneling decay and to a corresponding increase in the binding energy of the *A^+^*-state (compare curves 1 and 2 in Figure 2c).

### 3.2. Dependence of the Spectral Intensity for Recombination Radiation in a Quantum Dot with an Impurity Complex on the Parameters of the Surrounding Matrix (of the Heat-Bath)

Let us consider the process of radiative transition of an excited electron to a level of $A^{+}$-center. The energy structure of the considered system “quantum dot with impurity complex *A^+^ + e*” under conditions of dissipative tunneling is shown in Figure 1. The Coulomb interaction of an electron with a hole is accompanied by a radiative transition of an electron to an energy level of $A^{+}$-center with the radiation wavelength λ. Estimates have shown that, depending on the QD parameters and on the adiabatic potential of the electron, as well as on the temperature and on strength of the external electric field, the wavelength of the recombination radiation can vary in the range λ = 0.1–70 μm.

The energy spectrum of an electron in the size-quantized band, taking into account (10), is written in the form
(29)$$E_{n,l}=\frac{{\tilde{X}}_{n,\;l}^{2}E_{h}}{R_{0}^{*2}}+kT\ln\left[4sh\left(\frac{\Omega }{v_{LA}\sqrt{kT}}\right)sh\left(\frac{2\Omega }{v_{TA}\sqrt{kT}}\right)\right],$$
here ${\tilde{X}}_{n,l}$ is the root of the Bessel function of half-integer order l+1/2.

The wave function of an electron is given by an expression of the next form
(30)$$\Psi_{n,l,m}\left(r,\theta ,\varphi \right)=Y_{l,m}\left(\theta ,\varphi \right)\frac{J_{l+\frac{3}{2}}\left(K_{n,l}r^{*}\right)}{a_{h}^{\frac{3}{2}}\sqrt{2\pi }R_{0}^{*}\sqrt{r^{*}}J_{l+\frac{3}{2}}\left(K_{n,l}R_{0}^{*}\right)},$$
where $K_{n,l}$ is determined by the following expression
$$K_{n,l}=\sqrt{\frac{{\tilde{X}}_{n,\;l}^{2}}{R_{0}^{*2}}+\frac{kT}{E_{h}}\ln\left[4sh⟮\frac{\Omega }{v_{LA}\sqrt{kT}}⟯sh⟮\frac{2\Omega }{v_{TA}\sqrt{kT}}⟯\right]}$$

Spectral Intensity of Recombination Radiation (SIRR), taking into account the dispersion of QD sizes and the finite lifetime of the resonant $A^{+}$-state, is determined by an expression of the next form [24]
(31)$$\Phi \left(\omega \right)=\frac{4\omega^{2}\sqrt{\varepsilon }e^{2}}{c^{3}V}\left|\frac{P_{eh}e_{0}}{m_{0}}\right|\int \sum_{nlm}{\left|M\right|}^{2}P\left(u\right)\frac{\Gamma_{0}}{\frac{\hbar^{2}\Gamma_{0}^{2}}{4}+{\left(E_{nlm}-E_{\lambda h}-\hbar \;\omega \right)}^{2}}du,$$
where $m_{0}$ is the mass of a free electron; $P_{eh}$ is the matrix element of the momentum operator at the Bloch amplitudes of band carriers; ω—the frequency of the emitted electromagnetic wave with polarization $e_{0}$; V—is the QD volume; P(u)—the Lifshits—Slyozov function [25]:(32)$$P\left(u\right)=\{\begin{array}{l} \frac{3^{4}e\;u^{2}\exp\left[-1/\left(1-2\;u/3\right)\right]}{2^{\;\frac{5}{3}}{\left(u+3\right)}^{\frac{7}{3}}\;{\left(3/2-u\right)}^{\frac{11}{3}}},\;u<\frac{3}{2},\\ 0,\;\;\;\;\;\;\;u>\frac{3}{2}. \end{array}$$

The wave function of $A^{+}$-state, in the case of a central location of the $A^{+}$-center QD, has the next form (see (27)):(33)$$\begin{array}{ll} \Psi_{h}\left(r\right)= & -C\int_{0}^{\infty }dt\exp\left\{-\beta t⟮-\eta_{h}^{2}-\beta_{h}^{*}-\frac{x_{0}^{*2}}{a_{n}^{*4}}+i4\Gamma_{0}^{*}+\frac{kT}{E_{h}\beta }\ln\left[4sh⟮\frac{\Omega }{v_{LA}\sqrt{kT}}⟯sh⟮\frac{2\Omega }{v_{TA}\sqrt{kT}}⟯\right]+\frac{3}{2\beta }⟯\right\}\times \\ & {\left(1-\exp\left(-t\right)\right)}^{-\frac{3}{2}}\exp⟮-\frac{\left(1+e^{-2t}\right)}{\left(1-e^{-2t}\right)}\times \frac{r^{2}}{2a_{n}^{2}}⟯ \end{array}$$
where C is the normalization factor determined by an expression of the next form
(34)$$C={\left\{\frac{2\sqrt{\pi }}{{\left[\eta_{h}^{2}\left(T\right)\right]}^{2}}\frac{\Gamma ⟮1+\frac{\eta_{h}^{2}\left(T\right)}{2}⟯\cdot a_{n}^{3}}{\Gamma ⟮\frac{\eta_{h}^{2}\left(T\right)-1}{2}⟯}\left[\frac{\eta_{h}^{2}\left(T\right)}{2}⟮\Psi ⟮\frac{\eta_{h}^{2}\left(T\right)}{2}+1⟯-\Psi ⟮\frac{\eta_{h}^{2}\left(T\right)}{2}-\frac{1}{2}⟯-1⟯\right]\right\}}^{-\frac{1}{2}},$$
here $\eta_{\lambda h}^{2}\left(T\right)$—is determined by the dispersion Equation (28).

The matrix element of the radiative transition of an excited electron to the $A^{+}$-center level is given by the next expression
(35)$$M=i\;\lambda_{0}\sqrt{\frac{2\pi \alpha^{*}I_{0}}{\omega }}\;\left(E_{\;n,l,m}-E_{h}\right)\langle \Psi_{h}\left(r\right)\left|\;\left({\vec{e}}_{\lambda },\vec{r}\right)\;\right|\Psi_{n,l,m}\left(\rho ,\theta ,\varphi \right)\rangle .$$

Taking into account (29), (30), and (33), the matrix element of the radiative recombination transition of an electron from the ground state of the size-quantized band to the $A^{+}$-center level in the QD can be represented in the next form
(36)$$\begin{array}{ll} M= & \frac{a_{h}^{-1}2^{\;-\frac{3}{2}}\pi^{\;-\frac{5}{4}}\sqrt{3}\;\beta^{\frac{1}{2}}}{\sqrt{2\pi }R_{0}^{*4}J_{\frac{3}{2}}\left({\tilde{X}}_{n,1}\right)}[\frac{\Gamma ⟮1+\frac{\eta_{h}^{2}\left(T\right)}{2}⟯}{{\left[\eta_{h}^{2}\left(T\right)\right]}^{2}\Gamma ⟮\frac{\eta_{h}^{2}\left(T\right)-1}{2}⟯}\times \\ & {\left[\frac{\eta_{h}^{2}\left(T\right)}{2}⟮\Psi ⟮\frac{\eta_{h}^{2}\left(T\right)}{2}+1⟯-\Psi ⟮\frac{\eta_{h}^{2}\left(T\right)}{2}-\frac{1}{2}⟯-1⟯\right]\;]}^{\;-\;\frac{1}{2}}\int_{0}^{+\infty }\int_{-\pi }^{+\pi }\int_{0}^{2\pi }r^{*2}dr^{*}\cos\theta \sin\theta d\theta d\varphi \times \\ & \int_{0}^{\infty }dt\exp\left[-t⟮-\eta_{h}^{2}\beta -\beta \beta_{h}^{*}-\frac{x_{0}^{*2}\beta }{a_{n}^{*4}}+i4\beta \Gamma_{0}^{*}+\frac{kT\beta }{E_{h}}\ln\left[4sh⟮\frac{\Omega_{n}}{v_{LA}\sqrt{kT}}⟯sh⟮\frac{2\Omega_{n}}{v_{TA}\sqrt{kT}}⟯\right]+\frac{3}{2}⟯\right]\times \;\\ & {\left(1-\exp\left(-t\right)\right)}^{-\frac{3}{2}}\exp⟮-\frac{\left(1+e^{-2t}\right)}{\left(1-e^{-2t}\right)}\times \frac{r^{*2}}{2}⟯Y_{l,m}\left(\theta ,\varphi \right)\frac{J_{l+\frac{3}{2}}⟮\frac{{\tilde{X}}_{n,l}}{R_{0}^{*}}r^{*}⟯}{\sqrt{r^{*}}J_{l+\frac{3}{2}}\left({\tilde{X}}_{n,l}\right)}, \end{array}$$
where $R_{0}^{*}=R_{0}/a_{h}$—the QD radius.

Calculation of (36) leads to integrals, giving the selection rules for quantum numbers *m* and *l*:(37)$$\int_{0}^{2\pi }\exp\left(\mathrm{im}\varphi \right)d\varphi =\left\{ \begin{array}{l} 2\pi ,\;\mathrm{if}\;m=0\\ 0,\;\mathrm{if}\;m\neq 0, \end{array}\right.$$
(38)$$\int_{0}^{\pi }P_{l}\left(\cos\theta \right)\cos\theta \sin\theta d\theta =\left\{ \begin{array}{l} \frac{2}{3},\;\mathrm{if}\;l=1\\ 0,\;\mathrm{if}\;l\neq 1. \end{array}\right.$$

Thus, the radiative transition of an electron to the $A^{+}$-center level is possible only from states with the values of quantum numbers *l* = 1 and *m* = 0.

The remaining integral over the radial coordinate $r^{*}$ has the next form
(39)$$\int_{0}^{\infty }dr^{*}r^{*\frac{3}{2}}J_{l+\frac{3}{2}}⟮\frac{{\tilde{X}}_{n,l}r^{*}}{R_{0}^{*}}⟯\exp⟮-\frac{\left(1+e^{-2t}\right)}{2\left(1-e^{-2t}\right)}r^{*2}⟯=\sqrt{\frac{{\tilde{X}}_{n,l}}{R_{0}^{*}}}\exp⟮-\frac{1}{2}⟮\frac{1-e^{-2t}}{1+e^{-2t}}⟯{⟮\frac{{\tilde{X}}_{n,l}}{R_{0}^{*}}⟯}^{2})\times {⟮\frac{1-e^{-2t}}{1+e^{-2t}}⟯}^{\frac{3}{2}}.$$

Taking into account (37), (38), and (39), for the square of the matrix element (36) we have
(40)$$\begin{array}{ll} {\left|M\right|}^{2}= & \frac{\;\beta_{h}{\tilde{X}}_{n,1}}{2^{2}\pi^{\;\frac{5}{2}}a_{h}^{2}R_{0}^{*9}{\left|J_{\frac{3}{2}}\left({\tilde{X}}_{n,1}\right)J_{\frac{5}{2}}\left({\tilde{X}}_{n,1}\right)\right|}^{2}}|\frac{\Gamma ⟮1+\frac{\eta_{h}^{2}}{2}⟯}{{\left[\eta_{h}^{2}\right]}^{2}\Gamma ⟮\frac{\eta_{h}^{2}-1}{2}⟯}\times \\ & {\left(1-\exp\left(-t\right)\right)}^{-\frac{3}{2}}\exp⟮-\frac{\left(1+e^{-2t}\right)}{\left(1-e^{-2t}\right)}\times \frac{r^{*2}}{2}⟯Y_{l,m}\left(\theta ,\varphi \right)\frac{J_{l+\frac{3}{2}}⟮\frac{{\tilde{X}}_{n,l}}{R_{0}^{*}}r^{*}⟯}{\sqrt{r^{*}}J_{l+\frac{3}{2}}\left({\tilde{X}}_{n,l}\right)}\\ & \times |\int_{0}^{\infty }dt\exp\left[-t⟮-\eta_{h}^{2}\beta -\beta \beta_{h}^{*}-\frac{x_{0}^{*2}\beta }{a_{n}^{*4}}+i4\beta \Gamma_{0}^{*}+\frac{kT\beta }{E_{h}}\ln\left[4sh⟮\frac{\Omega }{v_{LA}\sqrt{kT}}⟯sh⟮\frac{2\Omega }{v_{TA}\sqrt{kT}}⟯\right]+\frac{3}{2}⟯\right]\times \\ & {{\left(1-\exp\left(-t\right)\right)}^{-\frac{3}{2}}\exp⟮-\frac{1}{2}⟮\frac{1-e^{-2t}}{1+e^{-2t}}⟯{⟮\frac{{\tilde{X}}_{n,l}}{R_{0}^{*}}⟯}^{2}⟯{⟮\frac{1-e^{-2t}}{1+e^{-2t}}⟯}^{\frac{3}{2}}|}^{2} \end{array}$$

Taking into account (32) and (40) for the spectral intensity of recombination radiation (SIRR) in QD (31), we can write
(41)$$\begin{array}{ll} \Phi \left(X,T\right)= & \Phi_{0}\times \frac{\;a_{h}^{4}\beta_{h}{\tilde{X}}_{n,1}X^{2}}{R_{0}^{*12}{\left|J_{\frac{3}{2}}\left({\tilde{X}}_{n,1}\right)J_{\frac{5}{2}}\left({\tilde{X}}_{n,1}\right)\right|}^{2}}\times \int_{0}^{\frac{3}{2}}duP\left(u\right)\times \\ & {\left|\frac{\Gamma ⟮1+\frac{\eta_{h}^{2}}{2}⟯}{{\left(\eta_{h}^{2}\right)}^{2}\Gamma ⟮\frac{\eta_{h}^{2}-1}{2}⟯}\left[\frac{\eta_{h}^{2}}{2}(\Psi ⟮\frac{\eta_{h}^{2}}{2}+1⟯-\Psi \left(\frac{\eta_{h}^{2}}{2}-\frac{1}{2}\right)-1⟯\right]\right|}^{-1}\times \\ & |\int_{0}^{\infty }dt\exp\left[-t⟮-\eta_{h}^{2}\beta -\beta \beta_{h}^{*}-\frac{x_{0}^{*2}\beta }{a_{n}^{*4}}+i4\beta \Gamma_{0}^{*}+\frac{kT\beta }{E_{h}}\ln\left[4sh⟮\frac{\Omega }{v_{LA}\sqrt{kT}}⟯sh⟮\frac{2\Omega }{v_{TA}\sqrt{kT}}⟯\right]+\frac{3}{2}⟯\right]\times \\ & {{\left(1-\exp\left(-t\right)\right)}^{-\frac{3}{2}}\exp⟮-\frac{1}{2}⟮\frac{1-e^{-2t}}{1+e^{-2t}}⟯{⟮\frac{{\tilde{X}}_{n,l}}{R_{0}^{*}}⟯}^{2}⟯{⟮\frac{1-e^{-2t}}{1+e^{-2t}}⟯}^{\frac{3}{2}}|}^{2}\times \\ & \frac{\Gamma_{0}^{*}}{\Gamma_{0}^{*2}+{⟮\frac{X_{n,1}^{2}}{R_{0}^{*2}}+\frac{kT}{E_{h}}\ln\left[4sh⟮\frac{\Omega }{v_{LA}\sqrt{kT}}⟯sh⟮\frac{2\Omega }{v_{TA}\sqrt{kT}}⟯\right]-\eta_{\lambda h}^{2}-X)}^{2}}, \end{array}$$
where $X=\hbar \omega /E_{h}$; $\Phi_{0}=\sqrt{\varepsilon }e^{2}\left|P_{eh}e_{0}\right|/4\pi^{\;\frac{5}{2}}\hbar^{3}c^{3}m_{0}$.

Figure 3 a–c shows the SIRR dependence on the magnitude of the external electric field $E_{0}$. It can be seen that the decrease in the SIRR value with increasing of $E_{0}$ is accompanied by “dips”, that appear at certain values of the external electric field strength and temperature. In [1] it is shown that variation of the strength of the external electric field can lead to transformation of the shape of the double-well oscillatory potential, which simulates the system “QD—surrounding matrix”, while the transition to the symmetric shape of the double-well oscillatory potential is accompanied by the appearance of a peak in the field dependence of tunneling probability (see inset in Figure 3a). Thus, the nature of the dip appears to be related to the effect of resonant tunneling, when the double-well oscillator potential becomes symmetric.

An increase in the SIRR value with the temperature increasing (compare curves 1 and 2 in Figure 3a) is associated with an increase in the overlap integral of the wave functions of the initial and final states due to temperature smearing of energy levels. It should be noted that the presence of dissipative tunneling makes the optics of quantum dots very sensitive to the parameters of the surrounding matrix, which determine, respectively, the frequency of the phonon mode (see Figure 3b,c) *ɛ*_L_ and the constant of interaction with the contact medium (with the heat-bath) *ɛ_C_*. With an increase in the value of *ɛ_L_*, the wave function of the $A^{+}$-state “spreads” due to the electron—phonon interaction, which is accompanied by a decrease in the SIRR value. An increase in the parameter *ɛ_C_* leads to an increase in the “viscosity” of the surrounding matrix, i.e., to a decrease in the probability of dissipative tunneling. As a result, the binding energy of the $A^{+}$-state increases, and the overlap integral of the wave functions of the initial and final states decreases, which leads to a decrease in the SIRR value.

When constructing the figures, the task was to show the features of the field dependence of the binding energy of the quasi-stationary state of the hole in the complex $A^{+}+e$
and the spectral intensity of recombination radiation (SIRR) associated with the dynamics of the temperature-dependent peak of the dissipative tunneling probability (see, for example, [1]). The choice of such parameter values is related, firstly, to the fact that InSb was chosen as the QD material; therefore, the parameters were expressed in Bohr units: the effective Bohr radius $a_{d}\approx 65$ nm and the effective Bohr energy $E_{d}\approx 0.001$ eV. Secondly, the choice of the numerical values of the parameters in Bohr units was carried out by numerical studies of the possibility of identifying the indicated features of the curves.

The high sensitivity of the spectra of recombination radiation and dissipative tunneling to the strength of an external electric field inspires a certain optimism for using the considered system “*A^+^ + e*” (a hole localized on a neutral acceptor interacting with an electron localized in the ground state of a quantum dot) for the diagnosis of amino acids and other ligands conjugated or surrounding QDs.

## 4. Discussion and Conclusions

Within the framework of the zero-radius potential model in the adiabatic approximation in combination with the one-instanton method, an analytical solution of the problem for bound states of a hole localized at a neutral acceptor interacting with an electron localized in the ground state of a spherically symmetric quantum dot in the presence of dissipative tunneling in an external electric field has been obtained. The solution of this problem made it possible to calculate in the dipole approximation the field dependence of the intensity of recombination radiation associated with the optical transition of an electron from the ground state of a quantum dot to the quasi-stationary state of the *A^+^*- center. It is shown that the field dependence of both the binding energy of the *A^+^*- state and the recombination radiation contains “dips” that appear at certain values of the external electric field strength and temperature. The “dips” are associated with the transformation of the shape of the double-well oscillatory potential, which models the “quantum dot—surrounding matrix” system, while a significant increase in the probability of dissipative tunneling is interpreted as the effect of resonant tunneling. Studies of the curves revealed a rather high sensitivity of SIRR to such parameters of the surrounding matrix as temperature, the constant of interaction with the contact medium, the frequency of phonon modes, and the strength of the external electric field, which makes recombination radiation in the system under consideration attractive for obtaining information about the medium surrounding the quantum dot. Using the above SIRR calculations, one can obtain a formula for the dependence of the threshold value of the photon energy in recombination radiation on the magnitude of the external electric field.

The threshold value of the recombination radiation energy is defined as the sum
$${\left(\hbar \omega \right)}_{th}=E_{e}+E_{\lambda h}+E_{g}$$
where
$$E_{e}=\frac{{\tilde{X}}_{n,\;l}^{2}E_{h}}{R_{0}^{*2}}+kT\ln\left[4sh\left(\frac{\Omega }{v_{LA}\sqrt{kT}}\right)sh\left(\frac{2\Omega }{v_{TA}\sqrt{kT}}\right)\right]$$
−energy of the ground state level of an electron in a QD, taking into account its temperature dependence,$R_{0}^{*}=R_{0}/a_{h}$—the QD radius; ${\tilde{X}}_{n,\;1}$—root of the n-th order for Bessel function. $E_{\lambda h}$—the binding energy of the *A^+^*-center, determined by the dispersion Equation (28), $E_{g}$—band gap in a bulk semiconductor.

Then, in Bohr units, the threshold value of the recombination radiation energy can be written as
(42)$$X_{th}=\frac{{\tilde{X}}_{1,1}^{2}}{R_{0}^{*2}}+\frac{kT}{E_{h}}\ln\left[4sh\left(\frac{\Omega }{v_{LA}\sqrt{kT}}\right)sh\left(\frac{2\Omega }{v_{TA}\sqrt{kT}}\right)\right]+\frac{E_{g}}{E_{h}}+\eta_{\lambda h}^{2},,$$
where $X_{th}={\left(\hbar \omega \right)}_{th}/E_{h}$, Ω− parameter depending on the deformation potential of the QD material;$v_{LA},v_{TA}-$ velocities of longitudinal and transverse phonons; $\eta_{\lambda h}^{2}=E_{\lambda h}/E_{h}$; $a_{h},E_{h}-$ the Bohr radius and Bohr energy, respectively. For QD based on InSb at T=100 K; $U_{0}=0.3$ eV;$R_{0}=70$ nm; $\varepsilon_{L}=0.5$; $\varepsilon_{C}=1$, change in electric field strength in the range from 0 to $E_{0}=10^{4}$ V/cm leads to a shift in the threshold value of the energy of recombination radiation calculated by formula (42) and the SIRR maximum by approximately 30 meV.

Thus, in the presence of dissipative tunneling in an external electric field, the “quantum dot–impurity complex” system investigated in this work can be used to diagnose amino acids. Indeed, amino acids can be negatively or positively charged, they can interact with quantum dots, modifying their energy spectrum and impurity states and hence the spectrum of recombination radiation. It should be noted that earlier in [6], the question of the advantages of using semiconductor quantum dots for the study and diagnostics of biological systems has been discussed. In this case, the process of tunneling between the core-shell quantum dots and a biological object was not taken into account by the authors of [6].

As our studies have shown, the parameters of the surrounding matrix can have a significant effect on the value of SIRR due to the change in the probability of dissipative tunneling. This is important because it becomes possible to obtain additional information about biological objects.

## Figures and Tables

**Figure 1 sensors-22-01300-f001:**
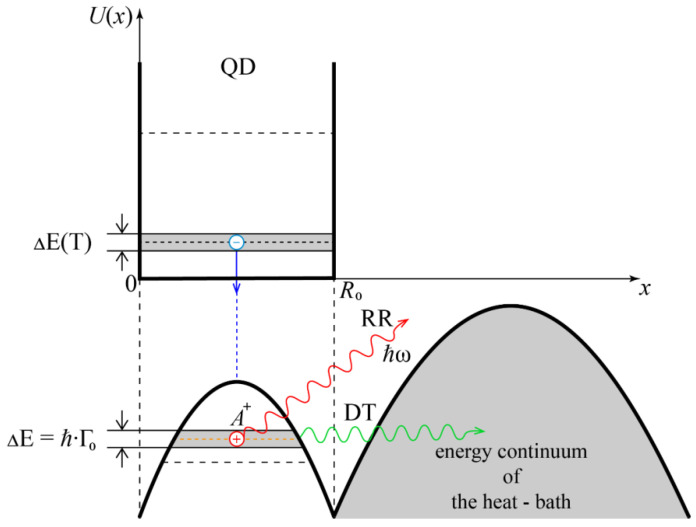
Energy structure of considered system based on the “quantum dot—impurity complex *A^+^ + e*” in the presence of dissipative tunneling. The electron is localized in the ground state of the quantum dot, and the hole is in the quasi-stationary state of the *A^+^*-center: ΔE(T)- is the magnitude of the temperature broadening of the QD electron energy level; ΔE = *ħ*Γ_0_—is the broadening of the impurity level associated with the tunneling decay of the *A*^+^-state; RR—recombination radiation; DT—dissipative tunneling.

**Figure 2 sensors-22-01300-f002:**
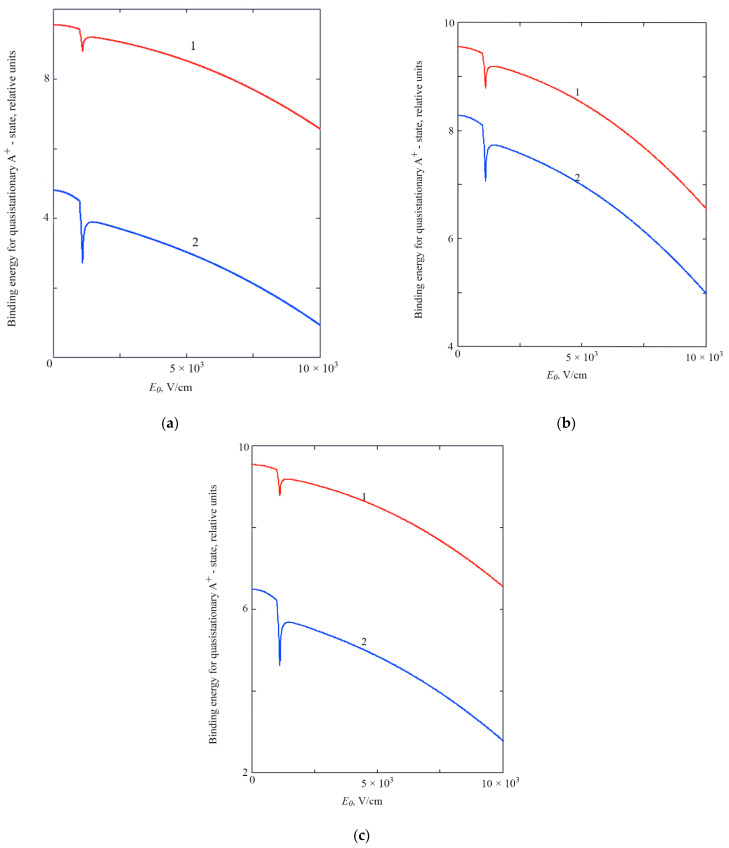
Dependence of the binding energy of the quasi-stationary state of a hole in a complex $A^{+}\;+\;e$ on the strength of the external electric field *E*_0_, at $R_{0}^{*}=1$; $U_{0}^{*}=300$: (**a**)—for different temperatures 1—T = 100 K; 2—T = 300 K; at $\varepsilon_{L}=0.5$; $\varepsilon_{C}=1\;$; (**b**)—for different values of the parameter $\varepsilon_{L}$ that determines the frequency of the phonon mode: 1—$\varepsilon_{L}=0.5$, 2—$\varepsilon_{L}=1$; at $\varepsilon_{C}=1\;$; T=100 K; (**c**)—for different values of the parameter $\varepsilon_{C}$ that determines the constant of interaction with the contact medium (with a thermostat or with the heat-bath): 1—$\varepsilon_{C}=1$, 2—$\varepsilon_{C}=0.5$; at $\varepsilon_{L}=0.5$; T = 100 K.

**Figure 3 sensors-22-01300-f003:**
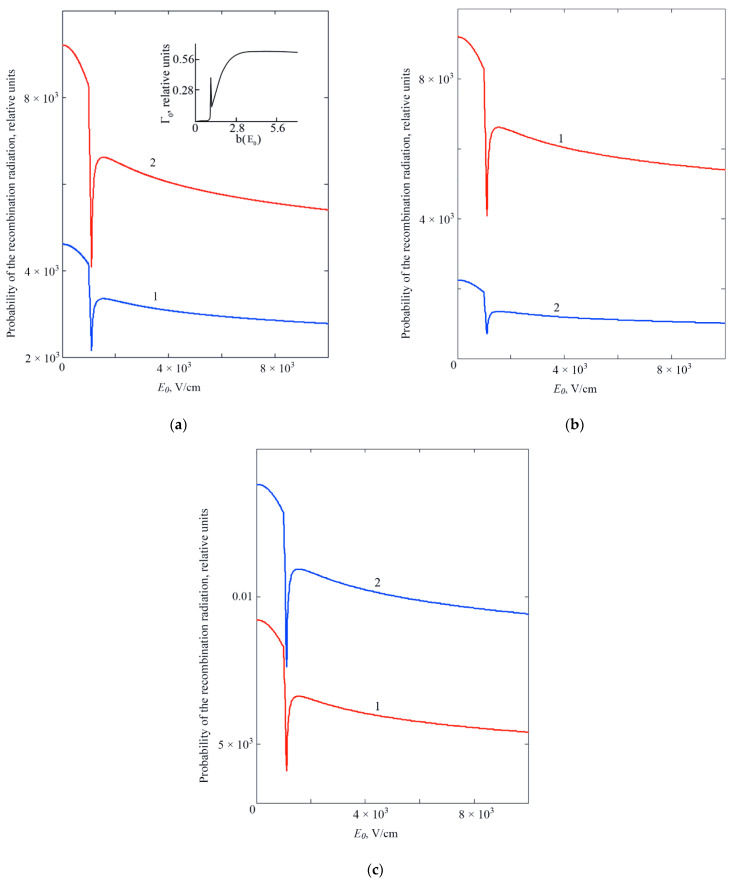
Dependence of the SIRR on the strength of the external electric field $E_{0}$ at ${\bar{R}}_{0}^{*}=1$;$U_{0}^{*}=300$; $\eta_{i}=4$: (**a**) for various values of temperature, *T,* K: 1—100, 2—300; $\varepsilon_{L}=0.5$; $\varepsilon_{C}=1$; (**b**)—for different values of the parameter $\varepsilon_{L}$ that determines frequency of the phonon mode: 1—0.5; 2—1; $\varepsilon_{C}=1$; T = 100 K; (**c**)—for different values of the parameter $\varepsilon_{C}$ that determines constant of interaction with the contact medium (with the heat-bath): 1—1.0; 2—0.5; $\varepsilon_{L}=1$, T = 100 K. The inset in Figure 3a shows dependence of the dissipative tunneling probability Γ_0_ on the parameter “b”, which determines the strength of the external electric field, obtained in [1].

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
