# Peer review of "Influence of an External Electric Field and Dissipative Tunneling on Recombination Radiation in Quantum Dots"

_sensors, 2022, doi:10.3390/s22041300_

Round 1

Reviewer 1 Report

The manuscript develops a physical model for a quantum dot system with impurities. The system is examined under the presence of an electric field and taking into account the presence of dissipative tunneling. Throughout their work the authors present an analytical formula that shows the relation of spectral intensity of recombination radiation (SIRR) to the quantum dot surrounding matrix such as the temperature, the electric field or the interaction of the contact medium. The work has merit and is adequate for publication, but first the authors must work on some points and improve their manuscript. I list them below:

  1. The main issue in the manuscript is the lack of in depth interpretation of the presented results. In sections 3.1 and 3.2 the authors develop systematically their analytical models. However, the discussion on their findings is very limited (1-2 paragraphs in the end of the section) and focuses mainly in describing the Figures.
  2. Similarly, the Discussion and Conclusion section is very small and does not include any Discussion. The strength of the manuscript will be highlighted by an elaborative final section that discusses the impact of the different parameters in the SIRR and also in the quantum dot sensor.
  3. The authors describe their model as one for a quantum sensor. A sensor as defined in their introduction (starting from line 32) is a system consisting of different components and not only the quantum dots. The analytical model they develop focuses on the quantum dot-impurity complex and its energetics. The authors should elaborate further on the relation of their models with the sensor and the impact of their findings to the sensor system.
  4. In Figures 1 and 3:
    1. For the plots, the authors choose some values for the temperature (T), the frequency of the phonon mode (εL) and the constant of interaction with the contact medium (εc). They must explain why do they use these specific values?
    2. For each plot only 2 values of T, εL, εc are used. This is a very small dataset. Wouldn’t also some plots that sweep these parameters at a constant electric field be useful? Especially in lines 282 – 284 the authors mention “The dip depth increases with increasing temperature … “ This is not obvious from the current graph. A plot showing the dip depth vs temperature will demonstrate this conclusion.
    3. For both Figures it is stated that there are “ dips at a fixed temperature” (lines 279 and 345-346). But in both Figure 1a and 2a the dips appear to be independent of the temperature. So, which is this fixed temperature that pronounces these dips? Can they plot this and explain the phenomenon?
    4. It will make it easier to the reader if the identification of each plot on the graph is marked with the actual parameter value instead of 1, 2. Or else a plot legend can be a solution.
    5. In the caption of Figure 1 there is information missing. Specifically, for (b) what is the T and εc and for (c) what are the values of εc chosen for each plot and what are the T and εL.
  5. Figure 2 gives a nice overview of the system that the authors work on. I believe it would facilitate the reader if it was the first Figure appearing already in the beginning of the manuscript. Even though the authors placed it there to focus on the radiative recombination, I found very useful looking at it already from the methods section.
  6. The term SIRR should be defined already in the first time that it appears on the manuscript, in line 317.
  7. In line 321 there is a mention to “P(u) - the Lipchitz – Slezov function”. Wouldn’t the correct naming be Lifshits – Slezov or LifshitsSlyozov ? I also have trouble to find the exact reference that the authors cite (ref. 25). I was aware of “The kinetics of precipitation from supersaturated solid solutions” and I found also the “On the kinetics of diffusion decomposition of supersaturated solid solutions” but the authors mention the word decay instead of decomposition. Maybe it is a translation error. Nevertheless the authors should make sure that the reference is correctly stated.
  8. When the authors discuss a contact medium they typically refer to a heat-bath. Can they comment if ligands surrounding the quantum dot can also be considered as the contact medium in their model?

Author Response

Let's update the list of these corrected positions:

  • An extensive APPENDIX is given at the end of the text of the article (pages 19 - 39 of the manuscript of the article and a reference to the literature [26] is added (lines 494-495 on page 18) (Reviewer's 2 request, remark 5))
  • On page 1 of the manuscript of the article (line 30) the link to the WEB page was removed (Reviewer's 3 request, remark 1)
  • On page 2 of the manuscript (lines 57-66), figure 2 in the Introduction is inserted and renumbered as “fig. 1” (request of Reviewer 1, remark 5). Accordingly, the original version of Figure 1 is renumbered as “Fig. 2" (p. 11 of the manuscript). Caption to fig. 1 (renumbered as fig. 2) has been changed in accordance with the request of Reviewer 1 (remark 4e).
  • On page 3 of the manuscript of the article (lines 74-84) a text insert was made in accordance with the request of Reviewer 1 (remark 3).
  • On page 6 (line 184) a correction was made in accordance with the request of Reviewer 3 (remark 3).
  • On page 6 (line 190) a correction was made in accordance with the request of Reviewer 3 (remark 4).
  • On page 7 (line 233) a reference is made to the APPENDIX (Reviewer's 2 request, remark 5).
  • On pages 11-12 of the manuscript (lines 311-317) a text insert was made in accordance with the request of Reviewer 1 (remark 1).
  • On page 12 of the manuscript, the term SIRR (spectral intensity of the recombination radiation) is given (line 335) in accordance with the request of Reviewer 1 (remark 6).
  • On page 12 of the manuscript (line 340), the name of the function "the Lifshits - Slyozov function" has been specified in accordance with the request of Reviewer 1 (remark 7). In accordance with the same remark, clarifications were made to the reference to the article [25] (lines 492-493, p. 18 of the manuscript).
  • A typo has been corrected in formula (38) on page 13 of the manuscript: the lower integration limit "-π" has been replaced by "0".
  • On page 16 of the manuscript (lines 393-397) a text insert was made in accordance with the request of Reviewer 1 (remark 1).
  • On pages 16-17 of the manuscript (lines 406-432) a text insert was made in accordance with the request of Reviewer 1 (remark 2).

We thank the Reviewers for a number of useful and important remarks.

Authors

Reviewer 2 Report

The paper “Physical model of a quantum sensor based on quantum dots for optoelectronics” reports the theoretical calculations of dissipative tunneling of a hole in quantum dot (QD) by using the instanton formalism. The authors say that the hole is as a results of impurity complex than interact with electron of QD. However, the authors need to explain the following matters.

1.- The QD is approaches some material in particular?

2.- The hole impurity is deep or shallow impurity type why originate this impurity?

3.- how many unit cells of the material crystal are contained in the QD?

4.- In your analytical approach used only the acoustic phonon of crystal why is not considered the optical phonon also?

5.- It was not clear how the bath vibrational modes were introduced in the developed analytical models.

6.- when you used the acoustical phonon, it was considered the phonon features of infinite crystal, however since you considered a QD structure the phonon also is affected by quantum size of structures, so the curves dispersion of phonons should be affected significatively also.

7.- It was not clear what was the principal contribution of the paper in the solid-state theoretical physic since there are many papers calculating the dissipative rate by the effect of external electrical polarization as you report in this paper.

8.- the details of the algebraic calculation procedures should be placed as an appendix. In the article, only the most relevant mathematical procedures for the discussion of the results should be pointed out.

9.- The article does not present any theoretical aspect aimed at possible application in sensors. The report points to possible charge transport mechanisms in the system due to an external polarization. In the case of sensors, energy transfer processes, interactions with the vibrational modes of analyte molecules would have to have been explored.

10.- The paper doesn´t have any results for quantum sensor, since in the any part of paper was discussed the single photon detection, single molecule detection together with the discussion of quantum limit related to quantum noise effect.

11.- The discussion is only in relation to reference [5], it is necessary to actualize the references. The references in the text should be written in the standard way as [ ].

Author Response

(The authors gave the same response as above.)

Reviewer 3 Report

  1. Please avoid citations in the brackets in Introduction. It ishould be limited.
  2. All materials need to be listed in Materials and methods. The source, type and manufacturer of quantum dots is missed.
  3. Line 168. Please change on “Larkin et.al.”
  4. Lines 173-177. If it is one reference, there is no need to refer twice in one sentence. Please, re-write.
  5. In the section Materials and Methods the conditions of analysis are also missed. A detailed measurement procedure should be given, including variations in environmental conditions - how the temperature, viscosity, etc. were changed, within what limits, on what devices it was measured.

Author Response

(The authors gave the same response as above.)

Round 2

Reviewer 1 Report

The authors present a reworked version of their manuscript in which they have addressed most of the reviewers’ comments. However, there are some points that must be improved before the work is accepted for publication.

  1. The main issue with the manuscript is stating that it develops a model of a quantum sensor, however it only focuses on the energetics of a single quantum dot. This has been already a remark from my part and from another reviewer during the first review round. The reply of the authors is summarized on lines 428-431, stating that experimental development was not the goal of this work. However, this reply indicates a wrong interpretation of the reviewers' comments and does not solve the issue, as no further theoretical modelling of a sensor system has been added.
    The manuscript title and abstract state that a model of a quantum sensor is presented, which is not correct. A sensor is a system containing multiple components and no such model is presented in this work. In my opinion, the correct interpretation of the work is that it presents a model of the energetics of a single quantum dot and discusses how this contributes towards the development of quantum sensors. I find the statements that the work describes a model of a  sensor misleading for the readers. I strongly suggest that the authors adjust the title, the abstract and the sentences stating that they model a sensor to an adequate form that reflects appropriately their work.
  2. In the first review round there have been questions from the reviewers on some details of the parameters used in their modelling (Reviewer 1 question 4a, Reviewer 2 questions 2 and 3). The authors answer successfully and in detail the questions, however they do not include their answers to the manuscript. It is highly possible that similar questions will be raised by the readers, so the clarifications the authors provided by these answers must be included in the manuscript or the appendix.
  3. In lines 73-83 the authors introduce a section to connect their work with the related applications. This addition is indeed of value, however there are 2 issues that must be addressed:
    1. For publication reference [6] “Zegrya G.G., Energy spectrum and lifetime of charge...”, I assume this is a translation issue. Do they author refer to this: https://doi.org/10.1134/S1063776109060016 ? In this added section the authors introduce the term open quantum dots, but this term does not appear in the reference I have found.
    2. The exact same paragraph is copied in the end of the manuscript, in lines 417-427. I suggest that the authors avoid this unnecessary text repetition and provide a conclusion paragraph that adds in the value of the manuscript, showing how their findings can assist in the development of future sensors.

Author Response

List of corrections (second iteration) 01/31/2022, (highlighted with a blue marker)

  1. The title of the article has been corrected (as requested by Reviewer 1): «Influence of an external electric field and dissipative tunneling on recombination radiation in quantum dots»
  2. We have corrected the text of Abstract (as requested by Reviewer 1)

Abstract. The effect of an external electric field and dissipative tunneling on the spectral intensity of recombination radiation in a quantum dot with an A+ + e impurity complex (a hole localized on a neutral acceptor interacting with an electron localized in the ground state of the quantum dot) is studied in the zero-radius potential model in the adiabatic approximation. The probability of dissipative tunneling of a hole is calculated in the one-instanton approximation. A high sensitivity of the recombination radiation intensity to the strength of the external electric field and to such parameters of the surrounding matrix (dissipative tunneling parameters) as temperature, the constant of interaction with the contact medium (or the heat - bath), and the frequency of phonon modes, has been revealed. It is shown that an external electric field leads to a shift of the recombination radiation threshold by several tens of meV, and a change in the parameters of dissipative tunneling has a noticeable effect on the spectral intensity of recombination radiation. It is shown that the resonant tunneling effect manifests itself in the form of "dips" in the field dependence of the spectral intensity of recombination radiation, which occur at certain values ​​of the external electric field strength and temperature. This opens up certain prospects for the use of the considered system “quantum dot – impurity complex A+ + e” under conditions of dissipative tunneling for the study and diagnostics of biological objects.

  1. We have corrected list of the key words.
  2. We have made some corrections to the text of the Introduction (highlighted with a blue marker), which are stylistic corrections.
  1. The terms "quantum sensors" were removed on the advice of Reviewer 1 and replaced by the term "considered systems".
  1. In section 2 “Materials and Methods” an addition was made from the answers to the questions of Reviewer 2 (during the first round of reviews) on the advice of Reviewer 1 and two references to the literature were added [11, 12]:

In semiconductor nanostructures, the concept of deep and shallow impurities is relative, since the depth of the impurity level depends on the characteristic size of the nanostructure. A+ - centers appear due to the attachment of an additional hole to a neutral acceptor, and the interaction potential of a hole with a neutral acceptor is not Coulomb, but short-range. Such centers have been found in quantum wells (GaAs/AlGaAs) [11,12 ].

As is known, the effective mass approximation is applicable if the exciton Bohr radius  is large compared to the crystal lattice constant . For a QD with a radius , the applicability condition for the effective mass method is that it () must exceed the  value by several orders of magnitude. It is easy to show that this criterion is satisfied by semiconductors with a small effective electron mass. Thus, for InSb-based QDs with effective masses of electrons  and holes , the exciton Bohr radius is , which is two orders of magnitude larger than the lattice constant ( ). In this work, calculations and plotting are performed for the radius of the QD , in this case  . Thus, the number of atomic layers in an InSb crystal turns out to be sufficient for the applicability of the effective mass method. The value of N given can serve as an estimate for the number of unit cells of the material crystals, which are contained in the QD.

  1. Small stylistic inserts are made on pages 4, 5, 7, 12.
  1. The reference [22] in the text after formula (10) is inserted instead of the erroneous reference [21].
  1. We have removed one redundant link on page 7 [1].
  1. On page 7, before formula (15), the incorrect reference [22] was corrected and replaced with [1].
  1. After formula (36), the clarification “the QD radius” has been added.
  1. After the caption to Figure 3, an insert was made on the advice of Reviewer 1 (from the answers to questions explaining the choice of parameters):

When constructing the figures, the task was to show the features of the field dependence of the binding energy of the quasi-stationary state of the hole in the complex  and the spectral intensity of recombination radiation (SIRR) associated with the dynamics of the temperature-dependent peak of the dissipative tunneling probability (see, for example, [1]). The choice of such parameter values ​​is related, firstly, to the fact that InSb was chosen as the QD material; therefore, the parameters were expressed in Bohr units: the effective Bohr radius nm and the effective Bohr energy eV. Secondly, the choice of the numerical values ​​of the parameters in Bohr units was carried out by numerical studies of the possibility of identifying the indicated features of the curves.

  1. We have removed paragraph in the end of chapter 3.1 (highlighted with a green marker).
  1. We have changed the text in the “Discussion and Conclusions” section on the advice of Reviewer 1:

Within the framework of the zero-radius potential model in the adiabatic approximation in combination with the one-instanton method, an analytical solution of the problem for bound states of a hole localized at a neutral acceptor interacting with an electron localized in the ground state of a spherically symmetric quantum dot in the presence of dissipative tunneling in an external electric field, has been obtained. The solution of this problem made it possible to calculate in the dipole approximation the field dependence of the intensity of recombination radiation associated with the optical transition of an electron from the ground state of a quantum dot to the quasi-stationary state of the A+- center. It is shown that the field dependence of both the binding energy of the A+- state and the recombination radiation contains "dips" that appear at certain values ​​of the external electric field strength and temperature. The "dips" are associated with the transformation of the shape of the double-well oscillatory potential, which models the "quantum dot - surrounding matrix" system, while a significant increase in the probability of dissipative tunneling is interpreted as the effect of resonant tunneling. Studies of the curves revealed a rather high sensitivity of SIRR to such parameters of the surrounding matrix as temperature, the constant of interaction with the contact medium, the frequency of phonon modes, and the strength of the external electric field, which makes recombination radiation in the system under consideration attractive for obtaining information about the medium surrounding the quantum dot. Using the above SIRR calculations, one can obtain a formula for the dependence of the threshold value of the photon energy in recombination radiation on the magnitude of the external electric field.

The threshold value of the recombination radiation energy is defined as the sum 

where

  • energy of the ground state level of an electron in a QD, taking into account its temperature dependence,the QD radius; root of the n-th order for Bessel function. the binding energy of the A+ - center, determined by the dispersion equation (28),  band gap in a bulk semiconductor.

   Then, in Bohr units, the threshold value of the recombination radiation energy can be written as

,

(42)

where ,  parameter depending on the deformation potential of the QD material;velocities of longitudinal and transverse phonons; ;  the Bohr radius and Bohr energy, respectively. For QD based on InSb at ; eV;nm; ;, change in electric field strength in the range from 0 to V/cm, leads to a shift in the threshold value of the energy of recombination radiation calculated by formula (42) and the SIRR maximum by approximately 30 meV.

Thus, in the presence of dissipative tunneling in an external electric field, the “quantum dot–impurity complex” system investigated in this work can be used to diagnose amino acids. Indeed, amino acids can be negatively or positively charged, they can interact with quantum dots, modifying their energy spectrum and impurity states, and hence the spectrum of recombination radiation. It should be noted that earlier in [6] question of the advantages of using semiconductor quantum dots for the study and diagnostics of biological systems has been discussed. In this case, the process of tunneling between the core-shell quantum dots and a biological object was not taken into account by the authors of [6].

As our studies have shown, the parameters of the surrounding matrix can have a significant effect on the value of SIRR due to the change in the probability of dissipative tunneling. This is important because it becomes possible to obtain additional information about biological objects.

  1. We have made corrections to the list of references, taking into account the inserts made, preserving the chronological sequence of references in the text of the article. We have changed reference [6] in accordance with the wishes of Reviewer 1.
  1. We have corrected individual references to literature in the APPENDIX.

Reviewer 2 Report

The authors answer all my successfully all my questions

Author Response

Thanks